# Silencing of *MNT1* and *PMT2* Shows the Importance of *O*-Linked Glycosylation During the *Sporothrix schenckii*–Host Interaction

**DOI:** 10.3390/jof11050352

**Published:** 2025-05-02

**Authors:** Manuela Gómez-Gaviria, José A. Martínez-Álvarez, Iván Martínez-Duncker, Andrea Regina de Souza Baptista, Héctor M. Mora-Montes

**Affiliations:** 1Departamento de Biología, División de Ciencias Naturales y Exactas, Campus Guanajuato, Universidad de Guanajuato, Noria Alta s/n, col. Noria Alta, C.P., Guanajuato 36050, Mexico; m.gomezgaviria@ugto.mx (M.G.-G.); martinezjose@ugto.mx (J.A.M.-Á.); 2Laboratorio de Glicobiología Humana y Diagnóstico Molecular, Centro de Investigación en Dinámica Celular, Instituto de Investigación en Ciencias Básicas y Aplicadas, Universidad Autónoma del Estado de Morelos, Cuernavaca 62209, Mexico; duncker@uaem.mx; 3Center for Microorganism’ Research, Biomedical Institute, Fluminense Federal University, Campus Valonguinho-Alameda Barros Terra, S/N, Niterói 24020-150, RJ, Brazil; andrearegina@id.uff.br

**Keywords:** biofilm formation, fungal cell wall, gene silencing, innate immune response, *O-*linked glycans, virulence, protein glycosylation

## Abstract

*Sporothrix schenckii* is a pathogenic fungus of worldwide distribution and one of the etiological agents of sporotrichosis. The cell wall is the first point of contact with host cells; therefore, its composition has been widely studied. It has a cell wall composed of chitin, β-glucans, and glycoproteins modified with *N-*linked and *O-*linked glycans. Protein *O-*linked glycosylation is mediated by two gene families, *PMT* and *MNT.* Therefore, we evaluated the relevance of protein *O-*linked glycosylation during the interaction of *S. schenckii* with the host. Independent silencing of the *MNT1* and *PMT2* was accomplished by interference RNA. Morphological analyses revealed defects in cell morphology in both yeast and mycelial cells; however, these defects differed between *MNT1* and *PMT2* silencing. Subsequently, the cell wall was characterized, and the silencing of these genes markedly changed cell wall organization. When the silenced strains interacted with human peripheral blood mononuclear cells, a reduced ability to stimulate the proinflammatory cytokines IL-6 and TNFα was found. However, the *PMT2-*silenced mutants also stimulated higher levels of IL-10 and IL-1β. Interaction with macrophages and neutrophils was also altered, with increased phagocytosis and decreased extracellular trap formation in both sets of silenced strains. Survival assays in *Galleria mellonella* larvae showed that silencing of any of these genes reduced the ability of *S. schenckii* to kill the host. In addition, the mutant strains showed defects in the adhesion to extracellular matrix proteins. These data indicate that *MNT1* and *PMT2* are relevant for cell wall synthesis and interaction with the host.

## 1. Introduction

Sporotrichosis is an acute or chronic mycosis caused by thermodimorphic fungi of the *Sporothrix* genus, which can affect humans and other mammals [1,2]. This mycosis has a worldwide distribution; however, it is more prevalent in tropical and subtropical areas, with Latin America having reported the most cases in recent decades [3,4]. *Sporothrix schenckii* is considered one of the leading etiological agents, presenting the widest geographical distribution. Currently, among the *Sporothrix* species, this is the one where the cell wall has been thoroughly analyzed. Similar to other fungal species, it is thought that its cell wall is essential for the viability and interaction with the environment. A model of the *S. schenckii* cell wall proposes that the innermost layer is composed of chitin, followed by β-glucans, which represent a major component in the cell wall and are linked by β-1,3-, β-1,4-, and β-1,6- bonds, and finally, in the outermost layer of the wall, there is a fibrillar layer known as peptidorhamnomannan (PRM) [5]. The *Sporothrix* cell wall also has a high content of glycoproteins, which are an important component and contribute to virulence and recognition by the host immune effectors [5,6]. These glycoproteins may be modified with *N*-linked, *O*-linked, and *C*-linked glycans or GPI anchors.

The *O*-linked glycosylation in fungi is a highly regulated process, which occurs mainly in the Golgi apparatus and is important for the function of various proteins [6]. This pathway has been widely studied in detail in *Saccharomyces cerevisiae* and *Candida albicans*. In the latter, *O*-linked glycans are linear oligosaccharides of one to seven α-1,2-mannose residues [7]. The addition of α-linked mannose residues to Ser/Thr residues begins in the lumen of the endoplasmic reticulum, where dolichol–phosphate–mannose (Dol-P-Man) is the sugar donor in a reaction catalyzed by any of the members of the protein *O*-mannosyltransferase (*PMT*) gene family [8,9,10]. Subsequently, glycoproteins are transported to the Golgi apparatus to begin the addition of other mannose residues by the action of Golgi α-1,2-mannosyltransferases, which are encoded by *MNT1* and *MNT2* [7,11]. These are GDP-mannose-dependent mannosyltransferases that may have redundant functions, but have a preference for adding particular mannose residues to *O*-linked glycans [11]. In *C. albicans*, Mnt1 adds the second mannose, while Mnt2 adds the third and fourth mannose units during the elongation step [7]. This biosynthetic pathway is similar in other fungal species, such as *Aspergillus fumigatus*, *Aspergillus nidulans*, and *Cryptococcus neoformans*, but the size of *PMT* and *MNT* gene families vary among species [12,13,14,15,16].

In *C. albicans*, the loss of *PMT2* leads to non-viable cells, indicating that this pathway is essential for cell growth and viability [15], but in the close relative *Candida tropicalis*, disruption of this gene did not compromise cell viability [17]. In addition, when *O*-linked glycans are incompletely elaborated, there are cell wall rearrangements, increased sensitivity to cell wall perturbing agents, defects in morphogenesis, and virulence attenuation [11,15,17]. Although *PMT2* disruption in *A. fumigatus* altered the cell wall, no changes in virulence were detected, in comparison to the wild-type strain [18]. Moreover, the *C. neoformans pmt2* null mutant did not show defects in cell growth or cell wall composition, but it had virulence attenuation [19]. Therefore, there is no obvious prediction of the *PMT2* contribution to *S. schenckii*’s biology. In the case of *MNT1*, in *C. albicans*, *A. fumigatus*, and *C. neoformans*, disruption of this gene led to defects in the cell wall composition and virulence attenuation [11,19,20,21]. It is worth noting that in these organisms, Mnt1 has a function in extending both *N*-linked and *O*-linked glycans during maturation within the Golgi apparatus [21,22]; therefore, the association of these phenotypes with exclusively shorter *O*-linked glycans is imprecise.

In *S. schenckii*, the *O*-linked glycosylation has not been widely studied; however, recent investigations have shown that when defects occur in the production of *N*-linked glycans, the *O*-linked glycan content in the cell wall is positively affected [23,24]. This could indicate that the glycosylation pathways in *S. schenckii* are highly dynamic and that they adapt when cells are facing environmental changes. It is also known that *N*-linked and *O*-linked glycans are representative components of the cell wall, with *N*-linked and *O*-linked glycans accounting for 45% and 55% of total cell wall glycan content [24]. The structure of *O*-linked glycans has been studied in PRM. The main glycans found in this complex are tetra and pentasaccharides composed of an α-1,2-mannobiose core added with one glucuronic acid unit that can be either mono- or dirhamnosylated [25].

Since the contribution of both *PMT2* and *MNT1* to fungal biology seems to be species-specific, here, we silenced *S. schenckii PMT2* and *MNT1* genes and characterized the phenotype of the mutant cells, focusing on the cell wall composition. Furthermore, we analyzed the impact of these silenced strains on the interaction with distinct human innate immune cells, and on virulence in the invertebrate model *Galleria mellonella*.

## 2. Materials and Methods

### 2.1. Microorganisms, Strains, and Culture Conditions

The microorganisms used in this study are shown in Table 1. The fungal wild-type (WT) strain was from ATCC (https://www.atcc.org, accessed on 30 April 2025) and the bacteria were from Invitrogen (https://www.thermofisher.com, accessed on 30 April 2025). Both *S. schenckii* silencing mutants were generated in this work.

The Luria–Bertani (LB) medium [1% (*w*/*v*) casein peptone, 0.5% (*w*/*v*) NaCl, 0.5% (*w*/*v*) yeast extract] was used for the selection and maintenance of *A. tumefaciens*. This was supplemented with 2% (*w*/*v*) bacteriological agar when a solid medium was required. In addition, when necessary, ampicillin (100 µg mL^−1^) (GOLDBIO, St Louis, MO, USA) or kanamycin (150 µg mL^−1^) (GOLDBIO) was included in the medium. The *A. tumefaciens* growth was performed by incubating at 28 °C and shaking at 120 rpm. For the growing and selection of *E. coli* strains, LB broth plus ampicillin (100 µg mL^−1^) was used, and cells were incubated at 37 °C for 16 h and 120 rpm orbital shaking. A YPD medium [1% (*w*/*v*) yeast extract, 2% (*w*/*v*) gelatin peptone, and 3% (*w*/*v*) glucose] was used for the *S. schenckii* growth and propagation. The mycelial phase was obtained in YPD broth, pH 4.5 at 28 °C, for 4 days with orbital shaking at 120 rpm. Yeasts were obtained in YPD broth, pH 7.8, incubated at 37 °C for 4 days, and with orbital shaking at 120 rpm [24]. The fungal cells that had the binary vector were selected on YPD plates, pH 4.5, and the corresponding selection drug. For *PMT2*, we used hygromycin B (400 µg mL^−1^), and for *MNT1*, it was nourseothricin (25 µg mL^−1^). Yeast-like cell aggregates were disrupted by vortexing for 30 s or by incubating with 3 U β-glucanase from *Trichoderma longibrachiatum* for 1 h at 37 °C [26,27]. The former was used throughout the study for cell preparation.

### 2.2. PMT2 and MNT1 Silencing

For silencing, a 308 bp fragment of the *MNT1* open reading frame (ORF) was amplified with the primer pair 5′ CTCGAGCGACTCGTCCAGCGACCC 3′ and 5′ AAGCTTATCATCGACCGAGCGACTTCCC 3′ (underlined sequences were added recognition sites for *Xho*I and *Hind*III). This fragment was cloned into the pSilent-1 *Xho*I and *Hind*III sites [28], generating pSilent-1-*MNT1*-sense. To obtain the antisense fragment, a second pair of primers was used, which had the same sequence, but with adapters for *Stu*I and *Bgl*II. This was cloned into the corresponding sites of pSilent-1-*MNT1*-sense, generating the pSilent-1-*MNT1*-sense-antisense. For *PMT2* silencing, a 213 bp fragment of the ORF was amplified with the primer pair 5′ CTCGAGGCAGCTGTTGTCGAGACTGA 3′ and 5′ AAGCTTAGCAACAAGAAGCGAAATGG 3′ (underlined sequences were added recognition sites for *Xho*I and *Hind*III). This fragment was cloned into the pSilent-1 *Xho*I and *Hind*III sites, generating pSilent-1-*PMT2*-sense. For antisense, a second pair of primers with the same sequence was used; however, the adapter sequences were for *Bgl*II and *Kpn*I. This amplicon was cloned into the corresponding sites of pSilent-1-*PMT2*-sense, generating pSilent-1-*PMT2*-sense-antisense. Both pSilent-1-*MNT1*-sense-antisense and pSilent-1-*PMT2*-sense-antisense were used as templates to amplify a fragment of 2757 bp for *MNT1* and 2567 bp for *PMT2*, respectively, spanning from the promoter (*PtrpC*) to (*TtrpC*) of pSilent-1. The primer pair used for this reaction was 5′ CTGCAGATGCCAGTTGTTGTTCCCAGTGATC 3′ and 3′ GAGCTCCCTCTAAACAAGTGTACCTGTGCATT 5′ (underlined sequences correspond to adapters for *Sac*I and *Pst*I, respectively). The resulting amplicons for each gene were cloned into the *Sac*I and *Pst*I sites of the binary vector pCambia-Nou for *MNT1*, and pBGgHg for *PMT2* [24,29]. These constructs were used to transform *A. tumefaciens* AGL-1 and then *S. schenckii*, essentially as reported [30]. Monoconidial cultures were performed as previously described [30].

The insertion of the binary plasmid within the *S. schenckii* genome was confirmed by PCR, using primers amplifying a region of the gene conferring resistance to hygromycin B (5′ GGCGACCTCGTATTGGGAATC 3′ and 5′ CTATTCCTTTGCCCTCGGACGAG 3′) or the gene conferring resistance to nourseothricin (5′ TAAGAGAGGTCCGCAAGTAGATT 3′ and 5′ TTAGGGGGGCAGGCAGGGCATGC 3′) [24,31].

### 2.3. Analysis of Gene Expression and Insertional Events Within the Fungal Genome by qRT-PCR

Total RNA extraction and subsequent cDNA synthesis were performed as described [32], using an oligo (dT)20 primer. The RT-qPCR was performed on a Step One™ thermal cycler (Applied Biosystems, Waltham, MA, USA), using cDNA 200 ng µL^−1^ and following the instructions of the SYBR Green PCR Master Mix kit (Applied Biosystems). The relative expression levels were determined by calculating the 2^−ΔΔΔCt^ [33]. Expression data were normalized using the gene encoding the ribosomal L6 protein (primer pair used for amplification 5′ ATTGCGACATCAGAGAAGG 3′ and 3′ TCGACCTTCTTCTTGATGTTGTTGG 5′) [34]. For *MNT1*, the primer pair used was 5′ AGCTCAGTATTCGGCAGCAC 3′ and 3′ GCTTGTCGTCGTTGAGGAAGACC 5′, which amplified a 257 bp fragment, and for *PMT2*, 5′ ACACAGACACCCTCCGTCAA 3′ and 3′ GAGAACTTGCTGCTGTGCCTCGT 5′, which amplified a 263 bp fragment. The amplification efficiency of these primer pairs was 100%, 104%, and 98% for *MNT1*, *PMT2*, and L6, respectively.

To calculate the number of binary plasmid insertional events within the *S. schenckii* genome, a similar approach was used but genomic DNA was used as template instead [23,24,31].

### 2.4. Growth Curves and Microscopic Analysis of MNT1 and PMT2 Mutants

Cultures of each strain were grown in YPD, pH 7.8, at 37 °C for four days. The optical density at 600 nm (OD_600nm_) was adjusted to 0.2 with fresh medium and was further incubated at 37 °C, measuring the OD_600nm_ at 12 h intervals, for 72 h. To determine phenotype changes, hyphae and yeast-like cells were examined under brightfield microscopy, using a Zeiss Axioscope-40 microscope and an Axiocam MRc camera (both from Zeiss, Jena, Germany).

### 2.5. Cell Wall Analysis

Yeast-like cells were obtained from a 4-day culture, and were pelleted, washed three times with deionized water, and broken in a Braun homogenizer (Braun Biotech International GmbH, Melsungen, Germany) with 4 cycles of 3 min, with resting periods of 1 min on ice between cycles. To obtain the cell walls, they were centrifuged and subjected to a cleaning protocol with deionized water and NaCl, followed by incubation in boiled water with SDS, β-mercaptoethanol, EDTA, and Tris [35]. Subsequently, 5 mg of cell wall was hydrolyzed with 2M trifluoroacetic acid overnight. The acid was evaporated, this material was resuspended in deionized water, and the product was analyzed by high-performance anion-exchange chromatography with pulsed amperometric detection (HPAEC-PAD), as previously reported [35,36]. For protein quantification, walls were resuspended in 1N NaOH, boiled for 30 min, and neutralized with 1N HCl. The protein content was quantified using the Pierce BCA protein assay (Thermo-Fisher Scientific), as previously reported [35]. The exposure of β-1,3-glucan and chitin on the cell wall surface was also analyzed using fluorescently labeled lectins, as previously reported [37,38]. Cells were heat-killed (HK) by incubating at 60 °C for 2 h [36], and these were used to normalize labeling with lectins, as the fluorescence associated with HK cells was considered as 100% polysaccharide exposure at the cell wall surface.

### 2.6. Alcian Blue Binding Assay

To determine changes in net negative wall charge, yeast-like cells were stained with Alcian blue [39]. Four-day-grown yeast-like cells were pellet, washed with deionized water, and cell concentration was adjusted to DO6_00nm_ = 0.2. Aliquots containing one mL were pelleted, the supernatant discarded, and cells suspended in one mL of a 30 μg mL^−1^ Alcian blue solution in 0.02 M HCl (Sigma-Aldrich, San Luis, MO, USA) and incubated at room temperature for 10 min. Then, cells were centrifuged, the supernatant saved, and used to measure OD at 620 nm. The amount of dye bound to cells was calculated as reported [39].

### 2.7. Analysis of N-Linked and O-Linked Glycans

Aliquots containing 1 × 10^9^ cells mL^−1^ were used to remove cell wall glycans. For *N-*linked glycan trimming, cells were incubated with 25 U of endoglycosidase H (New England Biolabs, Ipswich, MA, USA) and were incubated for 18 h at 37 °C [40]. To remove *O-*linked glycans, the cell pellet was treated with 0.1 M NaOH and incubated overnight at room temperature with gentle agitation [7]. In both cases, cells were pellet, the supernatant was recovered, the pH neutralized, and were kept at −20 °C until use. Sugar quantification was performed by the phenol–sulfuric acid method and HPAEC-PAD as reported [41,42].

### 2.8. Adhesion Assays

Polystyrene microtiter plates (Maxisorp, Nunc, Sigma-Aldrich) were coated by passive adsorption overnight at 4 °C with 100 µL per well containing 1 µg of each extracellular matrix component (EMC) [43]. The plates were then washed with PBS containing 0.05% (*v*/*v*) Tween 20. Non-specific binding was blocked by incubating the plates for 2 h at 37 °C with 1% (*w*/*v*) bovine serum albumin in PBS. After an additional washing step with PBS-Tween, 1 × 10^7^ yeasts were added per well, followed by incubation for 1 h at 37 °C. Plates were then washed to remove unattached cells, and 100 µL of anti-*S. schenckii* Hsp60 antibody (1:3000) was added to each well [44]. Plates were incubated for 1 h at 37 °C, washed with PBS-Tween, and then incubated with a peroxidase-conjugated goat anti-rabbit IgG antibody (1:4000 in PBS-Tween). Plates were washed with PBS-Tween and the reaction was developed with o-phenylenediamine substrate [0.5 mg mL^−1^ and 0.005% (*v*/*v*) H_2_O_2_ in 0.01 M sodium citrate buffer, pH 5.6]. The reaction was stopped after 5 min with 0.2 M H_2_SO_4,_ and the OD at 490 nm was measured using an automated plate reader. Each experiment was performed in triplicate. The ECM proteins assayed were bovine type II collagen (Sigma-Aldrich), human laminin, elastin, fibrinogen, recombinant fibronectin, recombinant thrombospondin-1, and type-I collagen (all from Sigma-Aldrich). ELISA-based assays showed that Hsp60 was similarly expressed at the cell surface of control and mutant strains.

### 2.9. Biofilm Formation

Biofilm formation was analyzed as previously reported [45]. Yeast-like cells were suspended in PBS, their concentration was adjusted to 1 × 10^7^ cells mL^−1,^ and 100 µL were placed in flat-bottom Nunc polystyrene 96-microtiter plates (Thermo Fisher Scientific, Waltham, MA, USA). Cells were incubated for 4 h at 30 °C to stimulate adhesion to the plastic surface, and then, wells were washed three times with PBS to remove non-adherent cells, 100 μL RPMI-1640 medium supplemented with L-glutamine (Sigma-Aldrich) was added to each well, and plates were incubated for 24 h at 37 °C. The wells were washed five times with PBS, and 100 μL absolute methanol was added and incubated for 15 min at room temperature. Once the alcohol was removed, the plates were air-dried, 100 μL of 0.02% (*w*/*v*) crystal violet was added to each well, incubated for 20 min at room temperature, washed three times with deionized water, 150 μL of 33% (*v*/*v*) acetic acid was added per well, and the absorbance at 590 nm was measured.

### 2.10. Protease and Lipase Activity

Secreted protease activity was measured as described [46]. Yeast-like cells were incubated for 4 days at 37 °C in YPD, pH 7.8, the cells pellet by centrifuging, culture media dialyzed against PBS, and proteins were concentrated in an Amicon Ultra centrifugal filter with Ultracel-3K (Sigma-Aldrich). Aliquots containing 50 µg protein were added to microplates, then 300 μL 5.0% (*w*/*v*) BSA (Sigma-Aldrich) in 50 mM sodium citrate, pH 3.2, were added, and plates were incubated for 30 min at 37 °C. Then, 100 μL 2 M perchloric acid was added and incubated for 15 min at 4 °C. Plates were centrifuged to pellet-precipitated proteins, and the absorbance at 280 nm was measured from supernatants. Control wells with sodium citrate and perchloric acid were used to measure the basal absorbance at 280 nm. The change in absorbance between the test and control well per minute was defined as one enzyme unit. For lipase activity, 50 µg secreted protein was mixed with 100 μL of 40 mM 4-methylumbelliferyl palmitate (Sigma-Aldrich) in 50 mM MES-Tris buffer, pH 6.0, and plates were incubated for 30 min at 37 °C. Then, 200 μL 50 mM glycine-NaOH buffer, pH 11.0 was added, and the released 4-methylumbelliferone was measured in an LS-5B luminescence spectrofluorometer (Perkin- Elmer, Waltham, MA, USA) with excitation and emission set at 350 nm and 440 nm, respectively [47]. One nmole 4-methylumbelliferone per min was defined as one enzyme unit. Intracellular enzyme activities were measured with the same methodologies, but first, cell homogenates were prepared, as described in Section 2.5 [45].

### 2.11. Ethical Considerations

In this study, the use of human primary cells was approved by the Ethics Committee of the University of Guanajuato (CIBIUG-P52-2021). Human cells were collected from healthy adult volunteers after information about the study was provided and written informed consent was obtained. The procedures were carried out following the Declaration of Helsinki.

### 2.12. Human Peripheral Blood Mononuclear Cells Isolation and Cytokine Stimulation

Blood samples were collected from eight healthy volunteer donors by venipuncture into tubes containing EDTA, and mixed with Histopaque-1077 (Sigma-Aldrich). The cell suspension was subjected to density centrifugation as described [48]. The human peripheral blood mononuclear cell (PBMC)–yeast-like cell interactions were performed in sterile 96-well cell culture microtiter plates, containing 100 μL aliquots of 5 × 10^6^ PBMC mL^−1^ and 100 μL of 1 × 10^5^ yeast-like cells mL^−1^. The interactions were incubated for 24 h at 37 °C with 5% (*v*/*v*) CO_2_. Then, plates were centrifuged for 10 min at 874× *g* at 4 °C, and supernatants were collected and kept at –20 °C until used. TNF-α, IL-1β, IL-6, and IL-10 were quantified by sandwich ELISA, using standard ABTS ELISA Development kits (Peprotech, Cranbury, NJ, USA), following the manufacturer’s instructions. Mock wells, where only human PBMCs were incubated, were used as controls. To assess the contribution of some immune receptors during the *Sporothrix*-PBMC interaction, immune cells were preincubated for 1 h at 37 °C with 200 µg mL^−1^ laminarin (Sigma-Aldrich), 10 µg mL^−1^ anti-mannose receptor (MR) (Thermo-Fisher Scientific, MA5-44033), 10 µg mL^−1^ anti-TLR4 antibody (Santa Cruz Biotechnology, Dallas, TX, USA sc-293072), 10 µg mL^−1^ anti-TLR2 antibody (Thermo-Fisher Scientific, Waltham, MA, USA 16-9922-82), and 10 µg mL^−1^ anti-CD11b antibody (CR3, Thermo-Fisher Scientific, MA5-16528) [36,49]. All antibody solutions were supplemented with 5 µg mL^−1^ polymyxin B (Sigma-Aldrich), to have lipopolysaccharide-free preparations [50]. As controls, PBMCs were preincubated with isotype-matched antibodies before interacting with fungal cells. The antibodies used were IgG1 at a concentration of 10 µg mL^−1^ (Santa Cruz Biotechnology, Cat. No.sc-52003, to control experiments with anti-TLR4 and anti-MR antibodies), 10 µg mL^−1^ of IgG2ak antibody (Thermo-Fisher Scientific, 14-4724-85, to control experiments with anti-TLR2 antibody), and 10 µg mL−^1^ of IgG2 antibody (R&D, Minneapolis, MN, USA, Cat. No. MAB9794, to control experiments with the anti-CD11b antibody) [36,51].

### 2.13. Phagocytosis by Human Monocyte-Derived Macrophages

Human macrophages were obtained by incubating PBMCs with recombinant human granulocyte–macrophage colony-stimulating factor (Sigma-Aldrich), as reported [52]. Yeast-like cells were adjusted at 2 × 10^7^ cells mL^−1^ in PBS and labeled with 1 mg mL^−1^ acridine orange (Sigma-Aldrich). Cells were thoroughly washed with PBS to remove unbound dye, and cells were adjusted to 3 × 10^7^ yeast cells mL^−1^. Interactions were performed in 800 μL aliquots of DMEM (Sigma-Aldrich), in six-well plates with a macrophage–yeast ratio of 1:6. The plates were incubated for 2 h at 37 °C and 5% CO_2_ (*v*/*v*) [53]. Human cells were washed with cold PBS and resuspended in 1.25 mg mL^−1^ trypan blue before being analyzed by flow cytometry on a FACSCanto II system (Becton Dickinson, Franklin Lakes, NJ, USA), as described [53]. For each sample, 50,000 events were collected per sample. Fluorescent signals were obtained using the channels FL1 (green) and FL3 (red), previously compensated with human cells without any labeling [53]. Positive cells for the green channel were classified as in the early stage of phagocytosis, those positive for both channels in the intermediate stage of phagocytosis, while only positive cells for the red channel were grouped in the late stage of the phagocytic event [53].

### 2.14. Analysis of Neutrophils’ Extracellular Traps

Human granulocytes were isolated from human peripheral blood as described [54]. Aliquots containing 175 μL of human granulocytes at 4 × 10^7^ cells mL^−1^ in RPMI 1640 were placed in 96-well plates previously coated with 1% (*w*/*v*) bovine serum albumin, and incubated for 30 min at 37 °C and 5% (*v*/*v*) CO_2_. Then, 25 μL of yeast-like cells at 4 × 10^8^ cells mL^−1^ were added to the wells and further incubated for 4 h at 37 °C and 5% (*v*/*v*) CO_2_. Then, plates were centrifuged for 10 min at 1800× *g* and 4 °C, and the supernatant was saved and used to quantify nucleic acids by spectrophotometry at 260 nm in a NanoDrop One (Thermo Fisher Scientific). Interactions containing only human granulocytes and PBS were included as a control.

### 2.15. Analysis of Virulence

Virulence was analyzed in *Galleria mellonella* larvae, as previously described [55]. Animals were from an in-house colony already established and were fed on a diet based on corn bran and honey [55,56]. Animal groups contained 30 larvae, which were inoculated with the same fungal strain. Yeast-like cells were adjusted to 1 × 10^7^ cells mL^−1^, and 10 µL was injected in the last left pro-leg, with a Hamilton syringe and a 26-gauge needle [55]. Inoculated larvae were monitored daily for two weeks and were kept in Petri dishes at 37 °C during the observation period. Animal death was defined as extensive body melanization and loss of irritability to external stimuli. A control group inoculated with PBS was also included. Hemolymph was collected from dead animals or those alive at the end of the experiment, this was anticoagulated and used to quantify colony-forming units (CFUs), as reported [56]. Alternatively, groups of 10 larvae were inoculated and incubated at 37 °C. At 24 h post-inoculation, hemolymph was collected, anticoagulated, and used for hemocyte counting, cytotoxicity, and phenol oxidase activity [57]. Cytotoxicity and phenoloxidase activity were measured in cell-free hemolymph, using the Pierce LDH Cytotoxicity Assay (Thermo Fisher Scientific), and 20 mM 3,4-dihydroxyDL-phenylalanine (Sigma-Aldrich), respectively. The former was defined as the release of lactate dehydrogenase to the extracellular compartment [56].

### 2.16. Statistical Analyses

Statistical analyses were performed using GraphPad Prism 6 software. All data were assessed for normality using the Shapiro–Wilk test to determine whether the statistical tests to be used should be parametric or nonparametric. Cytokine stimulation and phagocytosis were performed in duplicate with samples from eight healthy donors, while the rest of the in vitro experiments were performed at least three times. Cytokine profiles and phagocytosis were analyzed with the Mann–Whitney U test. Survival experiments with *G. mellonella* larvae were performed with a total of 30 larvae per group, and data were analyzed using the log-rank test and are reported in Kaplan–Meier survival curves. Other results were analyzed with Student’s t test. Statistical significance in all experiments was set at *p* < 0.05. All data are represented with mean and standard deviation.

## 3. Results

### 3.1. Silencing of Sporothrix schenckii MNT1 and PMT2 and Morphology Abnormalities

*MNT1* was previously characterized [58]. This encodes for a Golgi-resident α-1,2-mannosyl transferase belonging to the *MNT1*/*KRE2* gene family, which in *S. schenckii* is composed of three members [59]. These enzymes are involved in the glycan maturation step within the Golgi complex, where the *N*-linked glycan core and the *O*-linked glycans are elongated with mannose residues [6,10,22]. Even though the family members have a wide range of acceptors, glycosylating both types of glycans, *S. schenckii MNT1* is the sole family member that elongates *O*-linked glycans [58,59]. The *PMT* gene family is composed of three putative members in *S. schenckii*: SPSK_08548, SPSK_05892, and SPSK_08628 (gene symbols retrieved from https://www.ncbi.nlm.nih.gov/ accessed on 22 February 2024). Using *C. albicans* and *S. cerevisiae* proteomic information, organisms where this gene family has been experimentally characterized [12,15], bioinformatics analyses indicated that SPSK_08548 is the putative ortholog of Pmt2, and has a similarity of 65.4% and 50% for *C. albicans* and *S. cerevisiae* proteins, respectively; SPSK_05892, the putative Pmt1 ortholog, has a similarity of 44.1% and 45% for *C. albicans* and *S. cerevisiae* proteins, respectively; and SPSK_08628, the ortholog for Pmt4, has a similarity of 45.7% and 49% with *C. albicans* and *S. cerevisiae* proteins, respectively. Thus, SPSK_08548 is referred to hereafter as *PMT2*. The encoded polypeptide contains 758 amino acids, a putative transmembrane domain, and three MIR domains with beta-trefoil fold, characteristic of *PMT* family members [12].

Both genes were silenced using the *A. tumefaciens*-mediated *Sporothrix* transformation. The binary plasmids pCambia-Nou and pBGgHg were used to silence *MNT1* and *PMT2*. Both vectors were previously used to silence genes in *S. schenckii* [23,29,31]. Transformed cells were selected in YPD supplemented with either 25 µg mL^−1^ nourseothricin or 400 µg mL^−1^ hygromycin B, and mononuclear cells were selected by monoconidial passages and induction of dimorphism [30]. Confirmation of the binary plasmid inserted within the fungal genome was performed by PCR, amplifying a fragment of the marker that confers resistance to the selective drug. As controls, the wild-type (WT) strain was transformed with the empty pCambia-Nou and pBGgHg vectors, and two randomly selected PCR-positive colonies were selected from each transformation (Table 1) and used to assess the contribution of binary plasmid to the phenotype shown by the silenced strains. After gene expression analysis, three *MNT1*-silenced strains were selected, HSS49, HSS50, and HSS51, which showed *MNT1* expression levels of 14 ± 2.3%, 8.0 ± 1.1%, and 4.0 ± 0.9%, respectively, when compared to the *MNT1* expression in the WT strain. The control strains HSS67 and HSS68 showed similar *MNT1* expression levels to the WT strain (98.6 ± 0.6% and 99.2.0 ± 0.7%, respectively). Even though we analyzed dozens of PCR-positive strains transformed with pBGgHg-*PMT2*, we could not find strains showing high levels of gene silencing. The strains with the highest *PMT2* silencing levels were HSS54, HSS55, and HSS56, with 34.0 ± 1.5%, 33.0 ± 1.1%, and 30.0 ± 1.5%, respectively. The control strains HSS39 and HSS40 showed similar *PMT2* expression levels to those observed in the WT strain (99.2% ± 0.6% and 98.7% ± 0.9%, respectively).

The vectors used in this work are not site-directed, and it is not possible to control the number of integrative events within the genome. Thus, the number of insertional events was analyzed by qPCR, using the same primer pair used in expression analysis (see Section 2). Since the primer pair aligns in the sense and antisense regions used to generate the silencing constructions, it is expected to amplify two copies of this region when one insertional event has occurred within the *S. schenckii* genome. The *S. schenckii* genome is haploid [60]; therefore, a third copy of this region is expected to be quantified if the insertional event was ectopic. The six silenced mutants and the control strains showed one insertion event within the genome (the copy numbers for the selected regions were 3.0 ± 0.2, 2.8 ± 0.2, 2.9 ± 0.3, 3.0 ± 0.4, 2.9 ± 0.2, 2.8 ± 0.4, 3.1 ± 0.1, 3.0 ± 0.4, 3.2 ± 0.2, and 2.9 ± 0.2, for strains HSS49, HSS50, HSS51, HSS54, HSS55, HSS56, HSS67, HSS68, HSS39, and HSS40, respectively).

The colony morphology for the two groups of silenced mutants did not show significant changes when compared to the WT control strain, but cell morphology was aberrant (Figure 1). WT yeast-like cells showed the typical cigar-shaped morphology and hyphae were long septated filaments with small vacuoles and small and rounded conidia (Figure 1). The control strains showed a similar morphology. The three *MNT1*-silenced strains showed morphological changes in both stages: yeast-like cells were rounded and swollen with vacuolation inside the cytoplasm, whereas hyphae showed vacuolization and conidia were more abundant and swollen (Figure 1). In the case of the *PMT2*-silenced strains, yeast-like cells were rounded, vacuolated, and swollen; whistly hyphae were shorter and showed more branching (Figure 1). In both mutant series, yeast-like cells tended to aggregate (Figure 1), and this phenotype disappeared when cells were sonicated or treated with β-glucanase. Since yeast-like cells are the parasitic form obtained from infected tissues [61], we continued with the characterization of this cell morphology. When the length and wide of yeast-like cells were measured, both parameters were altered for the two mutant groups: the *MNT1*-silenced mutants showed 2.3 ± 0.4 µm and 1.4 ± 0.2 µm in length and width, respectively; whereas the *PMT2*-silenced strains showed 2.0 ± 0.5 µm and 1.35 ± 0.3 µm in average for length and width, respectively. These parameters are significantly different when compared to those measured in WT and control strains (3.3 ± 0.1 µm and 0.7 ± 0.3 µm on average, respectively; *p* < 0.05 when compared both parameters to the two mutant groups). The WT and control strains showed similar duplication times (average 8.6 ± 1.5 h), but for both groups, the *MNT1-* and *PMT2*-silenced mutants showed increased duplication times (12.6 ± 2.3 h on average for *MNT1* mutants, and 16.0 ± 2.0 h in average for *PMT2* mutants, *p* < 0.05 in both cases).

### 3.2. Silencing of Sporothrix schenckii MNT1 or PMT2 Affected the Cell Wall Composition and Protein Glycosylation

Since glycoproteins are part of the cell wall, and the genes under study are part of the *O*-linked glycosylation pathway, we next analyzed the impact of their silencing on the cell wall composition. Upon cell disruption, the cell walls were isolated and acid-hydrolyzed to break down polysaccharides and oligosaccharides into monosaccharides [35]. This treatment releases glucose and N-acetylglucosamine from glucans and chitin, respectively, and mannose and rhamnose from *O*-linked and *N*-linked glycans linked to cell wall glycoproteins [36]. The WT and control strains (HSS67, HSS68, HSS39, and HSS40) showed a similar content of monosaccharides, where glucose was the most abundant monosaccharide followed by mannose, rhamnose, and N-acetylglucosamine (Figure 2A,B). For the case of *MNT1* silencing, the three mutant strains showed similar monosaccharide levels among them, and when compared with the WT or control strains, these contained lower N-acetylglucosamine and rhamnose levels (Figure 2A). Glucose levels were not affected, but mannose content was higher than the WT and control strains (Figure 2A). For the *PMT2*-silenced strains, N-acetylglucosamine, rhamnose, and mannose levels were lower than the WT and control strains, but glucose levels were higher (Figure 2B). The three *PMT2*-silenced strains did not show differences in the cell wall composition when compared among them. Next, we analyzed the distribution of the cell wall polysaccharides β-1,3-glucan and chitin, which are mainly located in the inner part of the cell wall, underneath glycoproteins [36]. We used bulky FITC-conjugated lectins that only have access to the polysaccharide located at the cell surface [37,38]. The WT and control strains showed similar levels of exposed β-1,3-glucan and chitin at the cell surface (Figure 2C,D). For *MNT1*-silenced strains, the three mutants did not show significant variations in β-1,3-glucan, but significantly lower chitin labeling (Figure 2C), which is in line with the N-acetylglucosamine content (Figure 2A). For the *PMT2*-silenced mutants, a similar trend was observed for chitin labeling (Figure 2D), which once again is in line with the monosaccharide content (Figure 2B). In addition, the three *PMT2*-silenced strains showed higher β-1,3-glucan content at the cell surface than the WT or control strains (Figure 2D).

The cell wall protein content was also quantified. For this purpose, walls were extensively washed with detergent, boiled, and treated with reductive agents to remove adsorbed or loosely attached proteins [35]. The WT and the four control strains showed similar cell wall protein content (Table 2), but the three *MNT1*-silenced strains and the three *PMT2*-silenced strains showed increased and similar levels of wall protein (Table 2). It was previously shown that the *S. schenckii* cell wall has a net-negative charge, as it can bind the cationic dye Alcian blue [24]. The WT and control strains showed similar ability to bind the dye, but for the case of the *MNT1*-silenced strains, the three mutants barely bound the dye (Table 2). The three *PMT2*-silenced strains also showed a reduction in the dye bound in the wall but this was not as severe as that observed in the *MNT1*-silenced strains. Collectively, these results suggest that the silencing of *MNT1* or *PMT2* affected the cell wall composition and organization.

Next, we analyzed the cell wall *O*-linked and *N*-linked glycan content. The walls were trimmed with either endoglycosidase H or β-eliminated to trim *N*-linked or *O*-linked glycans, respectively. The released glycans were quantified by HPAEC-PAD. WT and control strains (HSS67, HSS68, HSS39, and HSS40) showed similar levels of both types of glycans (Figure 3A,B). The three *MNT1*-silenced strains showed a significant reduction in *O*-linked glycans and an increment in *N*-linked glycans (Figure 3A). The three *PMT2*-silenced strains did not show changes in the *N*-linked glycan content, but a reduction in the *O*-linked glycan levels was observed (Figure 3B). These data suggest changes in the glycosylation pathways.

### 3.3. Silencing of Sporothrix schenckii MNT1 or PMT2 Affected Cell Adhesion, Biofilm Formation, and Secreted Protease and Lipase

The cell wall characterization of the mutants indicated changes in the wall composition and organization. Therefore, it is likely that biological functions associated with the cell wall, such as cell adhesion, biofilm formation, and hydrolytic enzymes [45,62] may be affected. The WT and control strains showed a similar adhesion profile to extracellular matrix components, having the highest binding ability to laminin and fibrinogen, followed by type-I and type-II collagen, elastin, and fibrinogen. This adhesion profile is similar to that previously reported for *S. schenckii* yeast-like cells [43,63,64] (Figure 4). Cells failed to adhere to thrombospondin-1, as previously documented for *S. schenckii* [43,63,64] (Figure 4). Control wells with no ECM component gave threshold readings, similar to those where thrombospondin was present (Figure 4). Both silenced mutant cells showed a similar defect in adhesion, with a significantly lower ability to bind to laminin, elastin, fibrinogen, fibronectin, and type-I and type-II collagens (Figure 4).

Next, the ability to form biofilms was investigated. We followed a well-standardized methodology to assess biofilm formation by biomass staining with crystal violet [40,65,66]. It is relevant to note that non-adherent cells were discarded after cell adhesion was stimulated, which ensures the establishment of biofilms [66]. The biomass within biofilms formed by the WT and control cells was similar among them, but those generated by the *MNT1-* and *PMT2-*silencing mutants were significantly higher and similar across them (Figure 5).

We also measured the levels of secreted protease and lipase activity of yeast-like cells. The secreted protease activity was similar among the WT and control strains (Table 3). This tended to be significantly lower in the *MNT1*-silenced mutants, and this was even lower in the *PMT2*-silenced mutants (Table 3). A similar trend was observed with the secreted lipase activity, with similar levels for the WT and control strains and lower levels for the mutant strains (Table 3). Once again, the *PMT2*-silenced strains showed lower enzyme levels than the *MNT1*-silenced strains (Table 3). The lower levels of secreted enzymes may be due to defects in the secretory pathway that transports these enzymes to the extracellular compartment. Thus, we measured the content of intracellular enzyme activities. For both protease and lipase, the WT, control, and *MNT1*-silenced strains showed similar levels of activity (Table 3). In the case of the *PMT2*-silenced mutants, both protease and lipase activities were significantly higher than the WT or control strains (Table 3). These data suggest that, at least in the *PMT2*-silenced mutant, the reduction in secreted hydrolytic activities may be linked to defects in their secretion to the extracellular compartment.

### 3.4. Human Innate Immune Cell–Sporothrix schenckii Interaction Is Affected by MNT1 or PMT2 Silencing

Next, we analyzed whether the *MNT1* or *PMT2* silencing affected the *S. schenckii* yeast-like cells’ interaction with human innate immune cells, particularly PBMCs. We measured the proinflammatory cytokines TNFα, IL-6, and IL-1β, and the anti-inflammatory cytokine IL-10, as these have shown to be good parameters to assess how cell wall changes affect the host–*Sporothrix* interaction [23,24,31,49,64,67]. The WT and control strains stimulated similar levels of the four cytokines (Figure 6 and Figure 7). In the case of the *MNT1*-silenced strains, these stimulated similar and reduced levels of TNFα and IL-6, whereas no changes were observed in the IL-1β and IL-10 stimulation. We repeated these experiments with β-eliminated cells, where *O*-linked glycans are chemically removed from the cell wall, and the results were similar to those generated with untreated cells, suggesting that the observations are, indeed, linked to the reduced *O*-linked glycan content (Figure 6). When the contribution of some pattern recognition receptors on this cell–cell interaction was analyzed, we found that TNFα and IL-6 production stimulated by the WT and control strains was dependent on recognition by dectin-1, complement receptor 3 (CR3), TLR2, and TLR4, while mannose receptor (MR) was dispensable for these two cytokines production (Figure 6 and Figure 7). However, MR was required for the TNFα and IL-6 stimulation by the *MNT1*-silenced strains (Figure 6). Different from the WT and control cells, the stimulation of these two cytokines by *MNT1*-silenced strains was not sensitive to the presence of anti-CR3 and anti-TLR4 antibodies, indicating that these receptors are not participating in cytokine stimulation by these mutant cells (Figure 6). The IL-1β production was dectin-1-dependent for WT, control, and *MNT1*-silenced strains, whereas IL-10 production was dectin-1- and MR-dependent. Control interactions with irrelevant isotype-matched antibodies gave similar results to interactions with specific antibodies.

For the *PMT2*-silenced mutants, TNFα and IL-6 stimulation was reduced, whereas IL-1β and IL-10 productions were significantly increased (Figure 7). Similarly to the *MNT1*-silenced mutants, β-elimination of the *PMT2*-silenced strains stimulated similar cytokine levels as the untreated cells (Figure 7). Both TNFα and IL-6 production were dependent on dectin-1 and TLR2, but contrasting with WT and control cells, CR3 and TLR4 were dispensable for these cytokines by the mutant strains (Figure 7). For IL-1β stimulation, this was dependent on dectin-1 for WT, control, and silencing strains (Figure 7); while dectin-1 and MR were required for IL-10 stimulation in a similar way for all the tested strains (Figure 7). Control interactions with irrelevant isotype-matched antibodies gave similar results to interactions with specific antibodies. Collectively, these data indicated that silencing of either *MNT1* or *PMT2* affected the PBMCs–*S. schenckii* interaction.

Next, we assessed the interaction of the silenced mutants with human monocyte-derived macrophages. We analyze the fungal uptake by cytometry, and depending on the fluorescent channel, cells can be grouped in the early, intermediate, and late stages of the phagocytic process [49,53]. After 2 hours of interaction, most of the human cells were in the late stage of phagocytosis when interacting with the WT strain, followed by cells in the intermediate and early stages (Figure 8). The control strains for both silenced genes showed similar uptake profiles to the WT strain (Figure 8A,B). Both the *MNT1*- and the *PMT2*-silenced strains showed an increment in the phagocytosis at the three different stages, having more *PMT2*-silenced cells phagocytosed than the *MNT1*-silenced mutants (Figure 8A,B). These results indicated that silencing of *MNT1* or *PMT2* positively affected the *S. schenckii* phagocytosis.

We also analyzed the interaction of *S. schenckii* yeast-like cells with human granulocytes, closely evaluating the ability of the silenced strains to stimulate neutrophil extracellular traps. Among the several components released by neutrophils during trap formation are the nucleic acids, which generate a sticky net that reduces the mobility of pathogens [68]. So, we measured the release of nucleic acids as an indirect parameter of the formation of extracellular traps. Control cells, with no fungal cells included, released low levels of nucleic acids, but this was significantly increased when human cells were incubated with WT cells (Figure 9A,B). The control strains for both silenced mutant sets showed similar ability to stimulate traps as the WT strain (Figure 9A,B). Both *MNT1*- and *PMT2*-silenced mutant strains showed a reduced ability to stimulate the extracellular traps when compared to the WT or control strains (Figure 9A,B). These data indicate that silencing of *MNT1* or *PMT2* also affected the interaction of *S. schenckii* with human granulocytes.

### 3.5. Virulence Is Attenuated in the Sporothrix schenckii MNT1- and PMT2-Silenced Mutans

Next, the virulence of the *MNT1*- and *PMT2*-silenced strains were analyzed in the alternative model of experimental sporotrichosis *G. mellonella*, since this generates similar results as the murine model of sporotrichosis, in terms of mortality rates [55,69,70,71,72]. The larvae infected with the WT and control strains showed a mortality rate of 80.9 ± 6.4%, with a median survival of 6.5 ± 0.9 days (Figure 10A,B). The larvae inoculated with *MNT1*-silenced strains showed similar survival curves, with a mortality rate of 12.2 ± 1.2% and a median survival of more than 15 days (Figure 10A). The *PMT2-*silenced mutants also showed virulence attenuation, generating a mortality rate of 6.7 ± 2.1% with a median survival of more than 15 days (Figure 10B). In both cases, the larvae infected with the silenced strains showed significantly different survival curves when compared to the WT or control strains (*p* < 0.05; Figure 10). The control group inoculated only with PBS did not show mortality during the two-week observation period (Figure 10). To associate larval death with the presence of fungal cells, hemolymph was collected, and the colony-forming units were quantified. A similar fungal burden was observed for the WT, control, and silenced strains (Table 4).

To further characterize the interaction between *S. schenckii* and *G. mellonella*, some hemolymph parameters were analyzed. Previous studies have shown that reduction in the hemocyte counts, cytotoxicity, melanin formation, and phenoloxidase activity are associated with virulence attenuation [23,31,57,64]. The larva groups inoculated with the WT or control strains showed similar levels in the four parameters under analysis (Table 4). However, both the *MNT1*- and the *PMT2*-silenced mutants showed significantly lower levels in the four investigated parameters when compared to either WT or control strains (Table 4). Collectively, these data indicate that silencing of either *MNT1* or *PMT2* negatively affected virulence in the model *G. mellonella.*

## 4. Discussion

*S. schenckii* belongs to the *Sporothrix* pathogenic clade and is the most studied species within this clade, especially in terms of its biology, genetics, and other fundamental aspects [61]. As in other pathogenic fungi, the cell wall plays an essential role in the interaction with the host. Therefore, it is relevant to analyze its synthesis, composition, organization, and dynamic changes [73,74]. The *N-* and *O-*glycans covalently bound to cell wall proteins play a crucial role in their structure and function, are species-specific, and show unique structures and compositions [8,75]. Such particularities generate distinctive molecular profiles that influence the interaction with the host immune response. In *S. schenckii*, the presence of these *N-* and *O-*glycans has been reported, most of them being part of the peptidorhamnomannan [25]. In addition, at least the *N*-linked glycans have been demonstrated to play a key role during the *Sporothrix*–host interaction [23,24,31,36].

In this study, we report the first silencing of two genes involved in the *S. schencki*i *O*-linked glycosylation pathway and evaluate its contribution to the biology of this organism and its role in pathogen–host interaction. To genetically manipulate this organism, we used the gene silencing strategy, which is well described, and although the insertion of the binary vector is random, several mutants with similar silencing levels were included to rule out the observed phenotypes being the product of insertional events [23,24,31,64]. The *PMT2* silencing in *S. schenckii* represented a greater difficulty compared to *MNT1*. In the case of *PMT2*, the maximum level of silencing obtained was 70%. Considering studies in other organisms, such as *C. neoformans* and *Schizosaccharomyces pombe* [76,77,78], it is likely that *PMT2* is an essential gene not only in the *O*-glycosylation pathway, but also in cell viability.

Silencing of *S. schenckii MNT1* or *PMT2* led to cell aggregation and relevant changes in growth and morphology. Since notable differences were observed between mutants of the different genes, it is possible to suggest that each of these genes contributes differently to the fungus’ biology. These results suggest that proper *O-*linked glycosylation is imperative for cell fitness and growth, as has been reported for mutants of these genes in *C. albicans*, *C. tropicalis*, *S. cerevisiae*, *S. pombe,* and *C. neoformans* [11,17,77,79,80,81]. It is hypothesized that the observed phenotypic changes could be related to alterations in the *S. schenckii* cell wall composition, as it has been demonstrated in *C. albicans* mutants in the *O-*linked glycosylation pathway [11,79]. In the case of *MNT1*-silenced mutants, a reduction in the percentage of N-acetylglucosamine and rhamnose was found, along with an increase in mannose, supporting this hypothesis. Furthermore, the results of Alcian blue binding assays and structural polysaccharide distribution analysis reinforce this idea. Different from the *MNT1*-silenced mutants, the *PMT2-*silenced strains showed an additional increase in glucan levels. These changes have been associated with the activation of regulatory mechanisms in response to stress, such as the cell wall integrity pathway [82,83]. This adaptive mechanism involves activation of the Pkc1-Mkc1 cascade and β-1,3-glucan synthase, which generates a compensatory increment in structural polysaccharides and their redistribution to avoid cell death [84]. Therefore, the increase in glucans observed in *PMT2*-silenced strains could be explained by the activation of the cell wall integrity pathway. This hypothesis is supported by the Alcian blue binding assay, in which the mutants showed a lower affinity for the dye, and the levels of polysaccharides exposed at the cell surface, which indicated modifications in protein glycosylation and cell wall rearrangement, respectively. Similar observations have been described in *S. schenckii OCH1* and *ROT2-*silenced mutants [23,24]. The increase in cell wall protein content is consistent with the compensatory response of *S. schenckii* to maintain cell wall integrity in the presence of *O*-linked glycosylation defects. Alternatively, the increment in cell wall protein could be related to alterations in the secretory pathways, which could generate an abnormal protein accumulation in the cell wall. However, this explanation is not consistent with the results of protease and lipase activity assays, which revealed a decrease in the enzyme secretion, accompanied by an increase in their intracellular activity. These findings suggest that deficiencies in glycosylation not only affect cell wall composition, but also protein secretion. This could impact several key processes, including cell adhesion and biofilm formation. Protein glycosylation and the secretory pathway are two intimately linked pathways. In *C. albicans* and *S. cerevisiae*, defects in the secretory pathway affected protein glycosylation, probably because local stress in the Golgi complex affects glycosyltransferases [85,86]. Similarly, defects in protein glycosylation affected protein secretion in *C. albicans* and *C. tropicalis* [35,45]. Therefore, the observed defect in protein secretion in the silenced mutants analyzed in this study is not surprising.

Both sets of silencing mutants exhibited a reduced ability to adhere to extracellular matrix components, but retained their ability to form biofilms, raising several possible explanations. One hypothesis is that, despite the reduction in adhesion to the extracellular matrix components, the cells might be able to form biofilms due to an increase in the production of exopolysaccharides or other extracellular components, which facilitates cell adhesion to each other and favors the formation of a cohesive structure [87]. However, this is unlikely given the already-mentioned defect in protein secretion. Alternatively, changes in the regulation of genes related to biofilm biogenesis could contribute to this phenomenon, suggesting that key regulators of gene expression are decorated with *O*-linked glycans.

As expected, a decrease in *O-*linked glycan content was observed in both *MNT1*-silenced and *PMT2*-silenced mutants. However, in the case of the latter, no compensatory effect of *N-*glycan content was observed, unlike what was found in *MNT1*-silenced mutants. This phenomenon is particularly interesting because this kind of compensatory mechanism has not been reported in other strains that are deficient in this gene. One possible explanation is that other members of the *MNT1* gene family may be upregulated in this genetic background. These gene products participate in the *N*-linked glycan extension [59]. This implies that in the *MNT1*-silenced mutants, the increment in *N*-linked glycans represents more sugar units per glycan rather than more glycosylated sites per protein.

*MNT1* and *PMT2* silencing resulted in a reduced ability to stimulate the cytokines IL-6 and TNFα when *S. schenckii* cells interacted with human PBMCs. However, *PMT2* silencing induced higher levels of IL-10 and IL-1β, comparing to both the *MNT1*-silenced and WT strains. This increased cytokine production is likely linked to the higher levels of β-1,3-glucan, as this stimulation was blocked by laminarin, an antagonist of dectin-1, and anti-TLR2 antibodies, the two main immune receptors for β-1,3-glucans [88,89]. The changes in the cell wall were translated into changes in the contribution of pattern recognition receptors when the mutant cells were interacting with human PBMCs, underscoring the close relationship between fungal cell wall structure and immune sensing.

Even though both sets of mutants were readily phagocytosed by macrophages, this was more pronounced in *PMT2*-silenced mutants. This may be explained by the increment in β-1,3-glucan content, as mutants with similar cell wall defects tended to be more phagocytosed by human monocyte-derived macrophages [23,31,64].

The fact that both the *MNT1*- and *PMT2*-silenced mutants poorly stimulated neutrophil extracellular traps, along with the poor ability to stimulate proinflammatory cytokines by human PBMCs make us hypothesize that fungal *O*-linked glycans are key for the establishment of a proinflammatory response in the host when interacting with the pathogen. This observation contrasts with that reported in *C. albicans*, where these cell wall structures are a minor contributor to the stimulation of proinflammatory effectors [90]. This discrepancy is likely explained by the structure of *O*-linked glycans in these species. While *C. albicans* has *O*-linked glycans composed entirely of mannose units, *S. schenckii* has *O*-linked glycans composed of mannose, glucuronic acid, and rhamnose [25]. Rhamnose is a rare monosaccharide in the kingdom of fungi that has proinflammatory properties when included in *S. schenckii* glycans [31].

*MNT1* and *PMT2* silencing led to virulence attenuation in the *G. mellonella* model. These observations are consistent with *MNT1* disruption in *C. albicans* and *A. fumigatus* and *PMT2* deletion in *C. albicans* [11,20]. Considering that similar fungal loads were recovered from all *MNT1* and *PMT2* silencing strains, it is unlikely that the results are biased by the inability of the silencing mutants to adapt and grow in host tissues. Instead, our findings suggest that the observed defects are due to alterations in the repertoire of virulence factors, especially those that depend on proper glycosylation to perform their molecular function. Among the virulence factors identified in *S. schenckii*, adhesins and biofilm formation appear to be especially susceptible to defects in protein glycosylation [31], as some of them are highly glycosylated proteins, containing mannose- and rhamnose-based oligosaccharides [23,44]. It was clear that the mutant strains recovered the ability to grow within larvae hemolymph because our in vitro characterization indicated growth defects. The host milieu is likely to be a more stressful scenario than the culturing medium, and adaptation to cope with different stresses is mandatory for cell survival. Then, it is possible to speculate that this adaptation process is behind the discrepancy with the in vitro growth. The lower cytotoxicity observed in larvae inoculated with the silencing strains suggests a reduction in their ability to induce cell damage, in line with lower virulence. In addition, the low-hemocyte, melanin, and phenoloxidase levels indicated that insect immunity did not require a major upregulation to control the fungal cells, suggesting an immunological tolerance because of the low levels of cell damage in the host.

In conclusion, this study demonstrates that *S. schenckii MNT1* and *PMT2* are key for proper *O*-glycosylation, as well as for the organization and composition of the cell wall. The silencing of these genes affected the interaction of the fungus with human PBMCs, macrophages, and neutrophils, along with its virulence. This study paves the way for future research on *O*-linked glycosylation as a drug target candidate to control the infection caused by *S. schenckii*.

## Figures and Tables

**Figure 1 jof-11-00352-f001:**
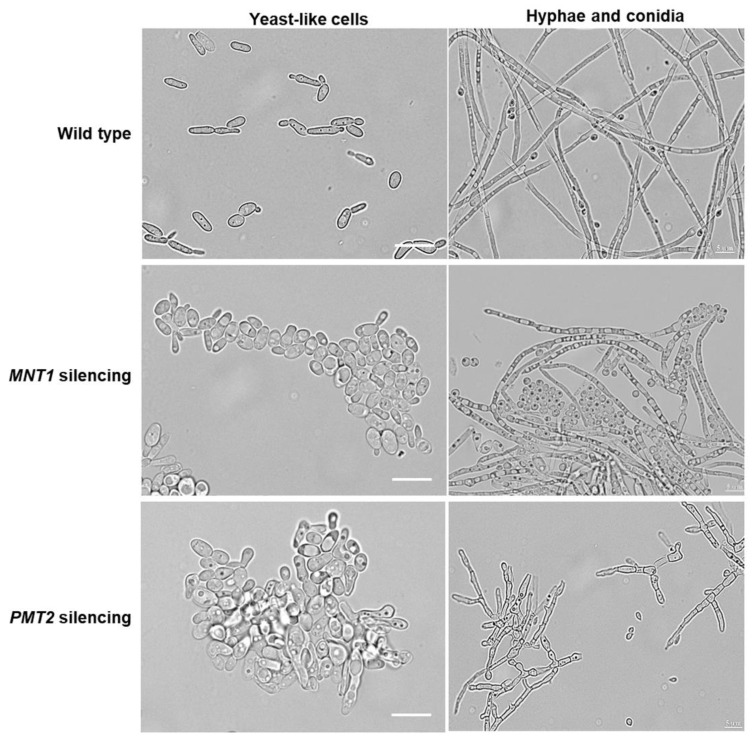
Cell morphology of *Sporothrix schenckii* wild-type, *MNT1*-silenced and *PMT2*-silenced strains. Bright-field microscopy of yeast-like cells, hyphae, and conidia. WT, strain 1099-18 ATCC MYA 4821. *MNT1* silencing, representative images of strains HSS49, HSS50, and HSS51. *PMT2* silencing, representative images of strains HSS54, HSS55, and HSS56.

**Figure 2 jof-11-00352-f002:**
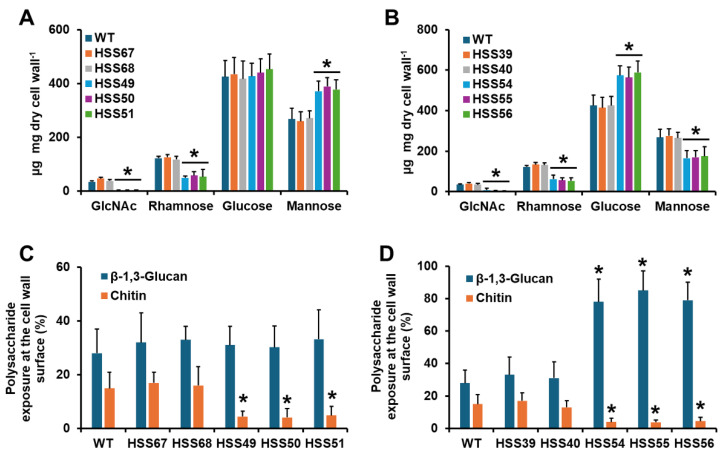
Cell wall analysis of *Sporothrix schenckii* wild-type, control, and silenced mutant strains. In (**A**,**B**), cells were disrupted, and walls were isolated and acid-hydrolyzed, breaking down oligosaccharides and polysaccharides into monosaccharides. These were separated and quantified by high-performance anion-exchange chromatography with pulsed amperometric detection. In (**A**), WT, control, and *MNT1*-silenced strains. In (**B**), WT, control, and *PMT2*-silenced strains. In (**C**,**D**), yeast-like cells were labeled with specific lectins for chitin (fluorescein isothiocyanate-conjugated wheat germ agglutinin) or β-1,3-glucan (IgG Fc-Dectin-1 chimera and anti-Fc IgG-fluorescein isothiocyanate), and the fluorescence associated with 300 cells was quantified. Data were normalized to the labeling obtained with heat-killed cells, which was considered 100%. In (**C**), WT, control, and *MNT1*-silenced strains. In (**D**), WT, control, and *PMT2*-silenced strains. For all panels, data are means ± SD of three biological replicates. Results were analyzed with Dunnett’s test and then the *T*-test. * *p* < 0.05 when compared to WT, HSS67, HSS68, HSS39 or HSS40 strains. WT strain was 1099-18 ATCC MYA 4821.

**Figure 3 jof-11-00352-f003:**
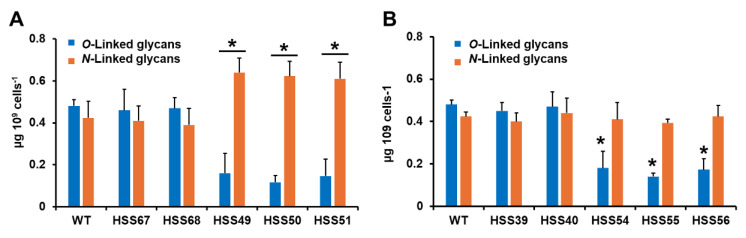
The cell wall *O*-linked and *N*-linked glycan content of *Sporothrix schenckii* wild-type, control, *MNT1*-silenced, and *PMT2*-silenced strains. In (**A**,**B**), yeast-like cells were incubated with endoglycosidase H or β-eliminated to remove *N*-linked or *O*-linked glycans, respectively. The released oligosaccharides were quantified by high-performance anion-exchange chromatography with pulsed amperometric detection and data normalized to 10^9^ yeast-like cells. In (**A**), WT, control, and *MNT1*-silenced strains. In (**B**), WT, control, and *PMT2*-silenced strains. Data are means ± SD of three biological replicates. Results were analyzed with Dunnett’s test and then the *T*-test. * *p* < 0.05 when compared to WT, HSS67, HSS68, HSS39, or HSS40 strains. WT strain was 1099-18 ATCC MYA 4821.

**Figure 4 jof-11-00352-f004:**
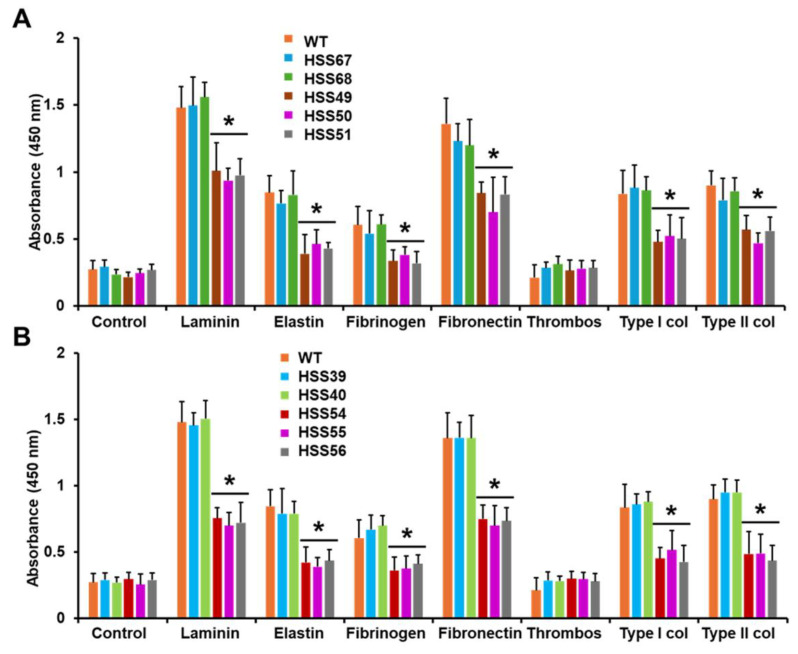
Adhesion of *Sporothrix schenckii* wild-type, control, *MNT1*-silenced, and *PMT2*-silenced strains to extracellular matrix components. In (**A**,**B**), microplates were coated with the extracellular component, and yeast-like cells were added and incubated for 1 h at 37 °C. Fungal cells were labeled with anti-*S. schenckii* Hsp60 antibodies, and then with peroxidase-conjugated goat anti-rabbit IgG antibody. Antibody presence was evidenced by adding hydrogen peroxide and o-phenylenediamine. Control, wells were coated only with bovine serum albumin. Thrombos, thrombospondin-1; Type I col, type-I collagen; Type II col; type-II collagen. Data are means ± SD of three biological replicates. Results were analyzed with Dunnett’s test and then the *T*-test. * *p* < 0.05 when compared to WT, HSS67, HSS68, HSS39 or HSS40 strains. WT strain was 1099-18 ATCC MYA 4821.

**Figure 5 jof-11-00352-f005:**
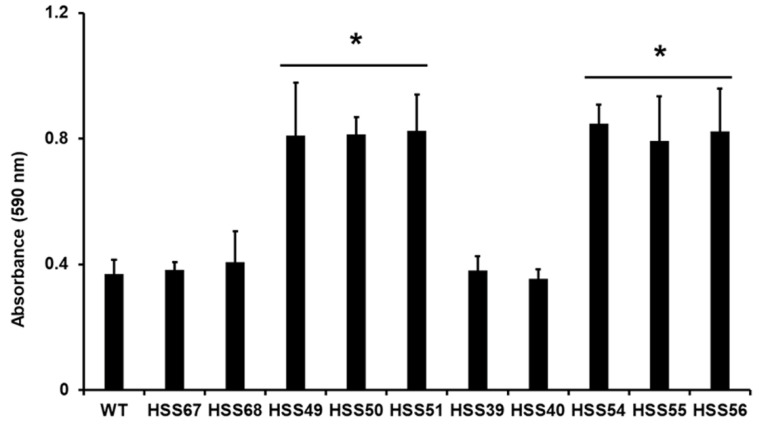
Biofilm formation of *Sporothrix schenckii* wild-type, control, *MNT1*-silenced, and *PMT2*-silenced strains. Microplates were coated with yeast-like cells and incubated to allow cell adhesion. Non-adherent cells were removed and biofilms were allowed to mature for 24 h at 37 °C. Cell biomass was estimated by crystal violet staining. Data are means ± SD of three biological replicates. Results were analyzed with Dunnett’s test and then the *T*-test. * *p* < 0.05 when compared to WT, HSS67, HSS68, HSS39, or HSS40 strains. WT strain was 1099-18 ATCC MYA 4821.

**Figure 6 jof-11-00352-f006:**
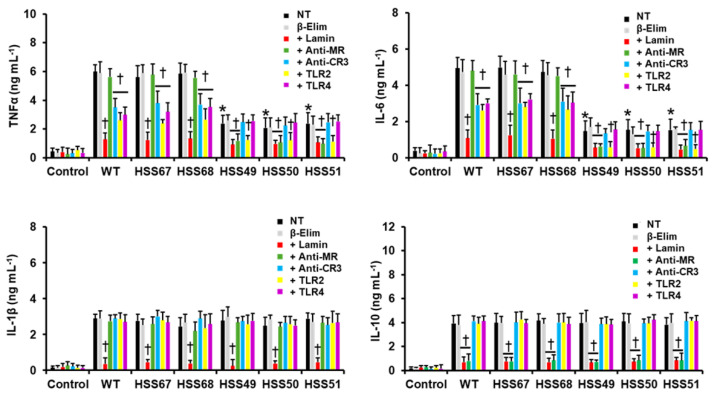
Cytokine production by human peripheral blood mononuclear cells stimulated by *Sporothrix schenckii* wild-type, control, and *MNT1*-silenced strains. Yeast-like cells and immune cells were coincubated for 24 h at 37 °C, and the supernatants were collected and used to quantify cytokines by ELISA. NT, no treatment applied to fungal cells; β-Elim, β-eliminated yeast-like cells; + Lamin, human cells preincubated with laminarin; + Anti-MR, human cells preincubated with anti-mannose receptor antibody; anti-CR3; human cells preincubated with anti-complement receptor 3 antibody; + anti-TLR2, human cells preincubated with anti-TLR4 antibody; and + Anti-TLR4, human cells preincubated with anti-TLR4 antibody. Data are means ± SD obtained with samples from eight donors, assayed in duplicate wells. Results were analyzed with Dunnett’s test and then the Mann–Whitney U-test. * *p* < 0.05 when compared to wild-type (WT) or control cells (HSS67 and HSS68). † *p* < 0.05 when compared to NT cells.

**Figure 7 jof-11-00352-f007:**
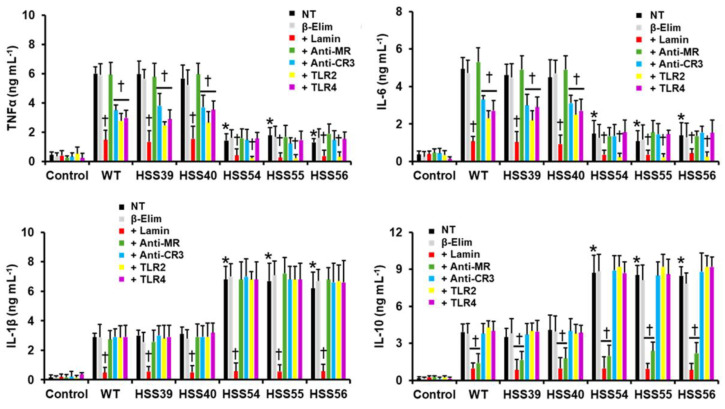
Cytokine production by human peripheral blood mononuclear cells stimulated by *Sporothrix schenckii* wild-type, control, and *PMT2*-silenced strains. Yeast-like cells and immune cells were coincubated for 24 h at 37 °C, and the supernatants were collected and used to quantify cytokines by ELISA. NT, no treatment applied to fungal cells; β-Elim, β-eliminated yeast-like cells; + Lamin, human cells preincubated with laminarin; + Anti-MR, human cells preincubated with anti-mannose receptor antibody; anti-CR3; human cells preincubated with anti-complement receptor 3 antibody; + anti-TLR2, human cells preincubated with anti-TLR4 antibody; and + Anti-TLR4, human cells preincubated with anti-TLR4 antibody. Data are means ± SD obtained with samples from eight donors, assayed in duplicate wells. Results were analyzed with Dunnett’s test and then the Mann–Whitney U-test. * *p* < 0.05 when compared to wild-type (WT) or control cells (HSS39 and HSS41). † *p* < 0.05 when compared to NT cells.

**Figure 8 jof-11-00352-f008:**
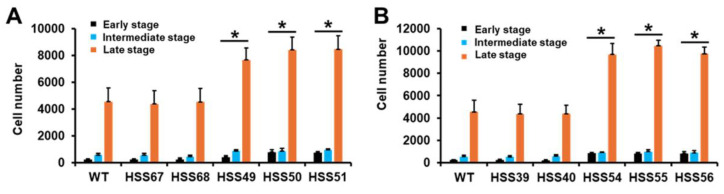
Analysis of *Sporothrix schenckii* phagocytosis by human monocyte-derived macrophages. Yeast-like cells and human cells were co-incubated for 2 h at 37 °C, and 5% (*v*/*v*) CO_2_ and phagocytosis were analyzed by flow cytometry. In (**A**), yeast-like cells of the *MNT1*-silenced strains were used. In (**B)**, yeast-like cells of the *PMT2*-silenced strains were used. Data are means ± SD obtained with samples from eight donors, assayed in duplicate wells. Results were analyzed with Dunnett’s test and then the Mann–Whitney U-test. * *p* < 0.05 when compared to wild-type (WT) or control cells (HSS67 and HSS68 in panel (**A**), or HSS39 and HSS41 in panel (**B**)).

**Figure 9 jof-11-00352-f009:**
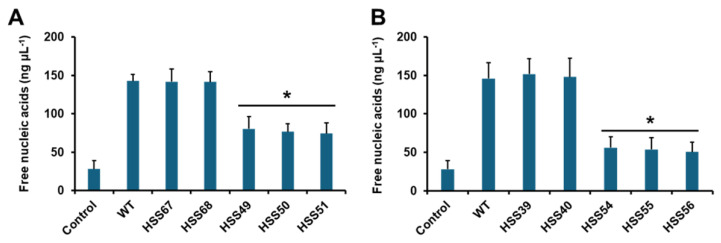
Stimulation of neutrophils extracellular traps by *Sporothrix schenckii* wild-type, control, *MNT1*-silenced, and *PMT2*-silenced strains. Yeast-like cells and human cells were co-incubated for 4 h at 37 °C and 5% (*v*/*v*) CO_2,_ plates were centrifuged, and supernatants were saved. The extracellular traps were measured by quantifying nucleic acids in supernatants. Control, human cells were incubated only with PBS. Data are means ± SD obtained with samples from eight donors, assayed in duplicate wells. Results were analyzed with Dunnett’s test and then the Mann–Whitney U-test. * *p* < 0.05 when compared to wild-type (WT) or control cells (HSS67 and HSS68 in panel (**A**), or HSS39 and HSS40 in panel (**B**)).

**Figure 10 jof-11-00352-f010:**
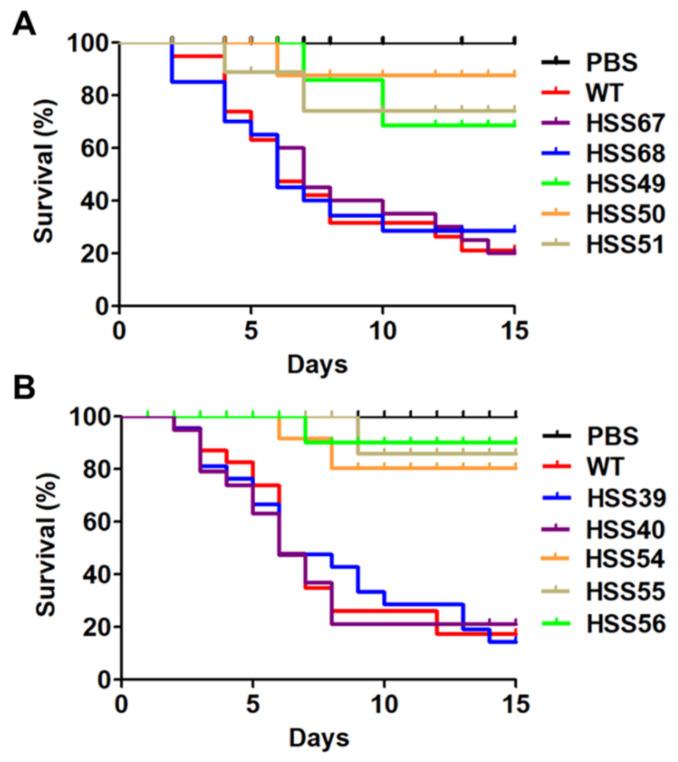
Virulence assays in *Galleria mellonella*. Experimental groups contained 30 larvae and were inoculated with the indicated *Sporothrix schenckii* strain, and mortality was recorded daily for two weeks. WT refers to the parental strain ATTCC MYA-4821. PBS, the control group that was inoculated only with phosphate-buffer saline (PBS). Data are shown in Kaplan–Meier plots. Panel (**A**) contains data on mortality associated with *MNT1*-silecenced mutants; while panel (**B**) contains results obtained with the *PMT2*-silenced mutants.

**Table 1 jof-11-00352-t001:** Microorganisms used and obtained in this study.

Microorganisms	Strain	Genotype
*Escherichia coli*	DH5α	*F- Φ80lacZΔM15 Δ(lacZYAargF)* *U169 recA1 endA1* *hsdR17 (rk-, mk+) phoA supE44* *λ-thi-1 gyrA96 relA1*
*Agrobacterium tumefaciens*	AGL1	AGL0 (C58 pTiBo542) recA::bla, T-region deleted Mop (+) Cb(R)
*S. schenckii*	1099-18 ATCC MYA 4821	Wild-Type
*S. schenckii*	HSS49 HSS50 HSS51	Strain 1099-18 ATCC MYA 4821 transformed with pCambia-Nou-*MNT1*
*S. schenckii*	HSS54 HSS55 HSS56	Strain 1099-18 ATCC MYA 4821 transformed with pBGgHg-*PMT2*
*S. schenckii*	HSS39 HSS40	Strain 1099-18 ATCC MYA 4821 transformed with pBGgHg
*S. schenckii*	HSS67 HSS68	Strain 1099-18 ATCC MYA 4821 transformed with pCambia-Nou

**Table 2 jof-11-00352-t002:** Cell wall protein content and ability to bind Alcian blue of *Sporothrix schenckii* wild-type, control, *MNT1*-silenced, and *PMT2*-silenced strains.

Strain	Cell Wall Protein Content (µg mg Cell Wall^−1^) *	Alcian Blue Bound (µg OD_600nm_ = 1.0^−1^) *
WT	186.5 ± 32.4	113.5 ± 12.5
HSS67	195.3 ± 28.4	118.1 ± 11.2
HSS68	188.6 ± 35.6	116.8 ± 19.2
HSS49	245.6 ± 48.2 †	6.5 ± 5.3 †
HSS50	239.8 ± 36.4 †	12.4 ± 7.9 †
HSS51	249.2 ± 41.1 †	9.8 ± 5.6 †
HSS39	188.5 ± 26.8	115.9 ± 21.5
HSS40	195.2 ± 35.5	110.5 ± 18.5
HSS54	268.5 ± 42.3 †	40.7 ± 21.4 †
HSS55	258.2 ± 39.5 †	32.4 ± 12.5 †
HSS56	271.3 ± 33.8 †	35.7 ± 17.7 †

* Data are means ± SD of three biological replicates.Results were analyzed with the Dunnett’s test and then the *T*-test. † *p* < 0.05 when compared with the values obtained with the WT or control strains. WT, 1099-18 ATCC MYA 4821.

**Table 3 jof-11-00352-t003:** Analysis of secreted protease activity, intracellular protease activity, secreted lipase activity, and intracellular lipase activity in *MNT1-* and *PMT2*-silenced strains.

Strain	Secreted Protease Activity (U) *	Intracellular Protease Activity (U)	Secreted Lipase Activity (U)	Intracellular Lipase Activity (U)
WT	1280.1 ± 258.8	3956.1 ± 358.6	412.5 ± 48.5	386.4 ± 68.5
HSS67	1145.6 ± 285.7	3845.3 ± 324.5.	435.6 ± 56.8	378.5 ± 56.8
HSS68	1205.5 ± 305.0	4102.5 ± 389.6	422.5 ± 42.1	401.0 ± 48.1
HSS49	656.2 ± 225.3 †	4258.2 ± 412.5	98.5 ± 36.5 †	423.4 ± 63.5
HSS50	708.4 ± 306.5 †	4125.3 ± 435.2	77.8 ± 45.5 †	389.7 ± 45.7
HSS51	777.5 ± 215.8 †	4356.1 ± 386.4	102.5 ± 48.7 †	405.2 ± 66.5
HSS39	1178.3 ± 296.7	4025.4 ± 401.5	435.6 ± 26.5	398.4 ± 78.9
HSS40	1258.4 ± 298.5	3953.5 ± 385.7	425.8 ± 29.8	412.4 ± 85.7
HSS54	325.1 ± 301.5 † ‡	5199.2 ± 356.2 † ‡	45.8 ± 45.6 † ‡	658.4 ± 96.4 † ‡
HSS55	268.1 ± 369.2 † ‡	5258.3 ± 478.5 † ‡	56.7 ± 52.4 † ‡	703.1 ± 55.5 † ‡
HSS56	298.5 ± 333.1 † ‡	5124.5 ± 402.8 † ‡	49.5 ± 47.8 † ‡	688.7 ± 88.4 † ‡

* Data are means ± SD of three biological replicates. For protease activity, one enzyme unit (U) was defined as ∆_280nm_ min^−1^. For lipase activity, one enzyme unit (U) was defined as one nmole 4-methylumbelliferone min^−1^. † Results were analyzed with the Dunnett’s test and then the *T*-test. *p* < 0.05 when compared with the values obtained with the WT or control strains. WT, 1099-18 ATCC MYA 4821. ‡ Results were analyzed with the Dunnett’s test and then the *T*-test. *p* < 0.05 when compared with the values obtained with strains HSS49, HSS50, or HSS51.

**Table 4 jof-11-00352-t004:** Fungal burden, cytotoxicity, hemocyte, melanin, and phenoloxidase levels in larvae of *Galleria mellonella* infected with *Sporothrix schenckii* wild-type, control, *MNT1-* or *PMT2*-silenced strains.

Strain	Colony-Forming Units (×10^5^) ^a^	Cytotoxicity (%) ^b^	Hemocytes (×10^6^) mL^−1^	Melanin ^c^	Phenoloxidase ^d^
PBS ^e^	0.0 ± 0.0	11.8 ± 3.4	3.4 ± 0.6	1.4 ± 0.8	0.5 ± 0.3
WT ^f^	3.4 ± 0.7	96.1 ± 8.9	8.0 ± 0.4	5.6 ± 0.8	3.9 ± 0.8
HSS67	3.4 ± 0.6	91.2 ± 6.6	7.7 ± 0.9	5.8 ± 0.4	3.9 ± 0.5
HSS68	3.1 ± 0.4	98.0 ± 9.7	7.9 ±0.7	5.9 ± 0.9	3.4 ± 0.9
HSS49	3.2 ± 0.8	22.4 ± 5.5 *	3.9 ± 0.6 *	2.2 ± 0.6 *	1.3 ± 0.4 *
HSS50	2.9 ± 1.0	30.5 ± 9.9 *	4.0 ± 0.6 *	2.4 ± 0.2 *	1.0 ± 0.3 *
HSS51	3.1± 0.8	22.1 ± 7.7 *	3.5 ± 0.8 *	1.9 ± 0.6 *	1.1 ± 0.9 *
HSS39	3.3 ± 0.8	94.9 ± 7.9	7.9 ± 0.4	5.3 ± 0.8	4.1 ± 0.9
HSS40	3.5 ± 0.5	97.5 ± 7.7	8.2 ± 0.9	5.4 ± 0.7	3.7 ± 0.8
HSS54	3.0 ± 0.9	18.4 ± 7.7 *	3.4 ± 0.5 *	1.5 ± 0.8 *	0.9 ± 0.3 *
HSS55	3.3 ± 0.7	12.5 ± 6.8 *	3.8 ± 0.7 *	1.6 ± 0.2 *	0.7 ± 0.2 *
HSS56	3.2 ± 0.9	17.5 ± 5.4 *	3.9 ± 0.9 *	1.9 ± 0.7 *	1.1 ± 0.8 *

^a^ Animals were decapitated and hemolymph was collected and used to calculate the colony-forming units by serial dilutions in YPD plates. ^b^ Lactate dehydrogenase activity was quantified in cell-free hemolymph. The 100% activity corresponds to data obtained with lysed hemocytes. ^c^ Calculated in the cell-free hemolymph as A_405nm._ ^d^ Enzyme activity defined as the Δ_490nm_ min^−1^ μg protein ^−1^. ^e^ Larvae inoculated only with PBS. ^f^ WT, strain 1099-18 ATCC MYA 4821. * *p* < 0.05 when compared with the values obtained in animals infected with the WT, or control strains.

## Data Availability

The original contributions presented in this study are included in the article. Further inquiries can be directed to the corresponding author(s).

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
