# Peer review of "Silencing of MNT1 and PMT2 Shows the Importance of O-Linked Glycosylation During the Sporothrix schenckii–Host Interaction"

_jof, 2025, doi:10.3390/jof11050352_

Round 1

Reviewer 1 Report

The authors studied the role of O-linked glycosylation in the biology of the fungal pathogen Sporothrix schenckii. They did that by characterizing knock down mutants for the PMT2 and MNT1 gene homologs, which potentially add the first and second/third mannose residues to the protein Ser/Thr by analogy with other fungal species. A third MNT2 gene has not been addressed in the present work. The authors analyzed 3 silenced mutants for each gene as compared with 2 empty vector-containing controls and the wild type.

In the mutants, the decrease in Mnt1 expression fell between 86 and 96%, while for PMT2 the maximum expression inhibition was 70%. The work is original for S. schenckii, it is clearly presented and in general the results support the conclusions. The features analyzed were: cell morphology, cell wall structure and composition, cell adhesion, biofilm formation, protease and lipase expression, stimulation of inflammatory mediators, and virulence in Galleria mellonella. Oligosaccharides can be involved in many fungal processes and, importantly, in the interaction with the host, specially when making part of glycoproteins that compose the cell wall. O-linked chains are less studied than N-linked sugars. Therefore,  the study brings relevant contribution to the fungal pathogen community. 

l. 378-380: please rephrase. This sentence is not clear. Is S. schenckii a family member? 

In the experiment shown in Figure 4, is there a control of the Hsp60 expression? If Hsp60 is somehow affected in the mutants, the interpretation of the results is compromised.

l. 623: do you mean "...strains stimulated similar levels..."? There are other similar sentences that could be improved, foe e.g., l. 734 "...the silenced strains showed significantly different survival curves." (the larvae infected by the the silenced strains showed...)

In the experiment of Figure 9,  measurement of the extracellular alone DNA is not enough to confirm the presence of cellular traps. Therefore, other experiments should be carried out or the conclusion should be softened.

l. 803-810: please specify the fungal species where these results have been described not to mix up with those found in S. schenckii.

l. 817-825: the authors are making conlusions about deffects in the secretion pathway based on the secretion of protease(s) and lipase(s). Maybe they could better ellaborate these conclusions, considering they have not tested genral secreted proteins. Have you estimated the amount of proteins in the supernatant before late growth phase? The problem could eventually be in the proteases and lipases measured. However, considering glycosylation is a part of the conventional secretion pathway, it would not be surprising to find general secretion issues with glycoproteisn bearing a signal peptide. Following these observations, how do you envision that gene regulators, that are more intracellular molecules supposedly without going through the secretory/glycosilation pathway, would be affected by O-glycosilation problems?

General comments: 

​1. Maybe the authors could restrict the number of citations for general statements, e.g., line 378, line 788, and several others​, considering this is not a review work.

2. In Figures 2, 3, 4, 8, 9, 10, please indicate within each figure which mutant it refers to in order to facilite anlysis by the reader without looking for A, B, ... in the legend.

3. Maybe the authors could restrict the description of some Materials and Methods that are less specific.

4. Please check the format of Table 2.

5. English reads really well, but some corrections are necessary.

Author Response

Reviewer1

We thank the Reviewer for his/her constructive comments

Major comments

The authors studied the role of O-linked glycosylation in the biology of the fungal pathogen Sporothrix schenckii. They did that by characterizing knock down mutants for the PMT2 and MNT1 gene homologs, which potentially add the first and second/third mannose residues to the protein Ser/Thr by analogy with other fungal species. A third MNT2 gene has not been addressed in the present work. The authors analyzed 3 silenced mutants for each gene as compared with 2 empty vector-containing controls and the wild type.

Reply: We thank the Reviewer for this succinct summary of our work. We would like to clarify that in S. schenckii, there is no putative ortholog for MNT2 as found in other fungal species. The family members are MNT1, KTR4, and KTR5, as described in the introduction. The last two participate in the N-linked glycosylation. No action was taken.

In the mutants, the decrease in Mnt1 expression fell between 86 and 96%, while for PMT2 the maximum expression inhibition was 70%. The work is original for S. schenckii, it is clearly presented and in general the results support the conclusions. The features analyzed were: cell morphology, cell wall structure and composition, cell adhesion, biofilm formation, protease and lipase expression, stimulation of inflammatory mediators, and virulence in Galleria mellonella. Oligosaccharides can be involved in many fungal processes and, importantly, in the interaction with the host, specially when making part of glycoproteins that compose the cell wall. O-linked chains are less studied than N-linked sugars. Therefore, the study brings relevant contribution to the fungal pathogen community.

Reply: We thank the Reviewer for this succinct summary of our work. No action was taken.

  1. 378-380: please rephrase. This sentence is not clear. Is S. schenckii a family member?

Reply: Comment addressed (see lines 378-379 of the revised manuscript).

¿In the experiment shown in Figure 4, is there a control of the Hsp60 expression? If Hsp60 is somehow affected in the mutants, the interpretation of the results is compromised.

Reply: We thank the Reviewer for this insightful comment. Before choosing this antibody, ELISA-based experiments were used to assess this aspect. Plastic plates were covered with yeast-like cells from the different mutants, then blocked with casein and used for ELISA with the anti-Hsp60 antibody as the primary antibody. No differences were observed in the binding ability of antibodies to the mutant and control cells, suggesting the Hsp60 levels across all tested strains are similar. We have added this clarification in lines 237-238. Please note that these antibodies have also been used in adhesion assays of other silenced strains generated in S. schenckii and S. brasiliensis, suggesting Hsp60 is constantly found in the Sporothrix cell wall (see https://pubmed.ncbi.nlm.nih.gov/39866864/; https://pubmed.ncbi.nlm.nih.gov/38786657/; and https://pubmed.ncbi.nlm.nih.gov/36422041/

  1. 623: do you mean "...strains stimulated similar levels..."? There are other similar sentences that could be improved, foe e.g., l. 734 "...the silenced strains showed significantly different survival curves." (the larvae infected by the the silenced strains showed...)

Reply: Comment addressed (see lines 626, 632,730, 731, 732, 735, and 736).

In the experiment of Figure 9, measurement of the extracellular alone DNA is not enough to confirm the presence of cellular traps. Therefore, other experiments should be carried out or the conclusion should be softened.

Reply: We appreciate the reviewer’s comment regarding the use of extracellular DNA quantification as a marker for the formation of extracellular traps (ETs). We acknowledge that extracellular DNA measurement alone does not definitively confirm the presence of ETs, as extracellular DNA can also originate from other processes such as cell lysis or apoptosis.

However, we would like to note that the quantification of extracellular DNA has been widely employed as an indirect but informative approach to assess ETs formation in various experimental models, particularly when evaluating responses to fungal pathogens. For example, Urban et al. (2006) and Branzk et al. (2014) used this strategy to investigate NET formation in response to Candida albicans, and other texts (https://www.nature.com/articles/ni.2987 , https://onlinelibrary.wiley.com/doi/10.1111/j.1462-5822.2005.00659.x, https://pmc.ncbi.nlm.nih.gov/articles/PMC10607474/ ).

That said, in response to the reviewer’s suggestion, we have revised the manuscript to soften our conclusion regarding ETs and clarified that our findings are based on indirect evidence (see line 719 of the revised version of the manuscript).

  1. 803-810: please specify the fungal species where these results have been described not to mix up with those found in S. schenckii.

Reply: Comment addressed (see lines 815-816 of the revised manuscript).

  1. 817-825: the authors are making conlusions about deffects in the secretion pathway based on the secretion of protease(s) and lipase(s). Maybe they could better ellaborate these conclusions, considering they have not tested genral secreted proteins. Have you estimated the amount of proteins in the supernatant before late growth phase? The problem could eventually be in the proteases and lipases measured. However, considering glycosylation is a part of the conventional secretion pathway, it would not be surprising to find general secretion issues with glycoproteisn bearing a signal peptide. Following these observations, how do you envision that gene regulators, that are more intracellular molecules supposedly without going through the secretory/glycosylation pathway, would be affected by O-glycosylation problems?

Reply: Comment addressed (see lines 840-845 of the revised manuscript).

General comments:

  1. ​Maybe the authors could restrict the number of citations for general statements, e.g., line 378, line 788, and several others​, considering this is not a review work.

Reply: We thank the reviewer for the comment regarding the number of citations used throughout the manuscript. We understand the concern, especially considering that this is not a review article. However, we intended to support each general statement and experimental interpretation with appropriate and relevant literature, ensuring that all perspectives considered in the construction of the manuscript are properly acknowledged. In particular, the cited works were selected to reflect different viewpoints and findings from the field, which we believe adds robustness to the discussion and places our results within a broader scientific context. Nevertheless, in response to the reviewer’s suggestion, we have carefully revised the manuscript and removed or consolidated some of the references in places where redundancy could be avoided, without compromising scientific accuracy.

  1. In Figures 2, 3, 4, 8, 9, 10, please indicate within each figure which mutant it refers to in order to facilite analysis by the reader without looking for A, B, ... in the legend.

Reply: We appreciate the reviewer’s suggestion to improve the clarity of the figures. As noted, all figures already include the specific names of the mutant strains analyzed. Additionally, Table 1 provides a detailed summary indicating the gene targeted in each mutant, which allows readers to easily identify which strain corresponds to which gene. We have checked the figures to ensure that the mutant names are displayed and consistent throughout the manuscript. While we understand the reviewer's comment about information between figures and legends, we believe the current format is sufficiently informative and avoids overloading the figures with excessive text. Including the full names made the legends big enough to have figures of a non-published size. The alternative to placing the full names in smaller cases made the legend unreadable. So, no action is taken at this point.

  1. Maybe the authors could restrict the description of some Materials and Methods that are less specific.

Reply: We appreciate the reviewer’s comment regarding the level of detail in the Materials and Methods section. However, we intentionally chose to provide a comprehensive description of the procedures to ensure that the experiments can be reproduced by other researchers. While some methods may be well established, including key details helps maintain transparency and reproducibility, which we consider essential, especially when working with biological systems that may vary depending on the specific conditions used. That said, we have reviewed the section to ensure that all content is relevant and appropriately concise, while still allowing replication of the study.

  1. Please check the format of Table 2.

Reply: Comment addressed.

  1. English reads really well, but some corrections are necessary.

Reply: Comment addressed.

Reviewer 2 Report

This study is an important contribution to the knowledge of the roll of genes synthesizing the O-glycoproteins of the cell wall of Sporothrix schenckii in immune response and outcome of the infection.

I do not any important suggestion for authors

Author Response

There are no comments to address.

Reviewer 3 Report

Manuela Gómez-Gaviria et al report the silencing of two genes involved in the S. schenckii O-linked glycosylation pathway and evaluate its contribution to the biology of this organism and its role in pathogen-host interaction. The silencing of MNT1 and PMT2 affected the interaction of S. schenckii with human PBMCs, macrophages, and neutrophils, along with its virulence.

  1. PMT2 has a low silencing efficiency (only 30-34%), but still significantly alters the phenotype. Does this suggest a key role for PMT2 in glycosylation? Is the residual activity of PMT2 sufficient to maintain partial function, or is there functional redundancy in other PMT family members? It needs to be further verified by gene knockout or overexpression assay.

2.Is the growth ability of silent plant affected? Does this affect its virulence against galleria mellonella?

3.The cell adhesion of silent strains decreased, but biofilm formation increased. Explain this phenomenon and discuss possible molecular mechanisms.

4.Is the survival of MNT1/PMT2 silencer in the host but reduced pathogenicity due to increased sensitivity to oxidative killing (e.g., ROS) in the host? The survival rate of the silenced strains under the conditions of active oxygen production by H₂O₂ or macrophages should be measured, and the expression of antioxidant genes (such as SOD and CAT) should be analyzed to determine whether the expression is regulated by glycosylation.

Author Response

We thank the Reviewer for his/her comments

Manuela Gómez-Gaviria et al report the silencing of two genes involved in the S. schenckii O-linked glycosylation pathway and evaluate its contribution to the biology of this organism and its role in pathogen-host interaction. The silencing of MNT1 and PMT2 affected the interaction of S. schenckii with human PBMCs, macrophages, and neutrophils, along with its virulence.

Reply: We thank the Reviewer for this succinct summary of our work.

PMT2 has a low silencing efficiency (only 30-34%), but still significantly alters the phenotype. Does this suggest a key role for PMT2 in glycosylation? Is the residual activity of PMT2 sufficient to maintain partial function, or is there functional redundancy in other PMT family members? It needs to be further verified by gene knockout or overexpression assay.

Reply: We would like to clarify that the highest silencing efficiency achieved for PMT2 was approximately 70%, not 30–34% as indicated. Despite this partial reduction in expression levels, we observed significant phenotypic alterations, which highlight the importance of PMT2 in S. schenckii. This is consistent with its proposed role as the initiating enzyme in the O-glycosylation pathway. Our findings suggest that PMT2 plays a key role in this process, and its partial silencing is sufficient to disturb important aspects of cell wall organization and fungal-host interactions. Although we cannot exclude the possibility of functional redundancy with other PMT family members, this remains an open question for further investigation, which right now is out of the scope of this study. Regarding the generation of null mutants, gene knockout strategies in Sporothrix remain challenging due to technical limitations in standardizing efficient homologous recombination in this genus. Nevertheless, obtaining a PMT2 knockout strain would indeed provide valuable insights, and we consider it a promising direction for future work, building upon this study. Having said that, we consider that our dataset is strong enough to sustain the conclusion drawn in our manuscript.

  1. Is the growth ability of silent plant affected? Does this affect its virulence against Galleria mellonella?

Reply: We would like to clarify that our study was performed using Sporothrix schenckii, a pathogenic fungus that affects humans and can grow on vegetal debris, not plants. We are not currently aware of any work where S. schenckii is grown in plants. We do not discard that this might be a relevant aspect to address, but it is out of the scope of our study.

  1. The cell adhesion of silent strains decreased, but biofilm formation increased. Explain this phenomenon and discuss possible molecular mechanisms.

Reply: The apparent phenomenon between decreased cell adhesion and increased biofilm formation in the silenced strains was indeed addressed in the Discussion section (lines 835-844).

In Sporothrix schenckii, the partial silencing of glycosylation-related genes may lead to changes in cell wall composition and surface architecture that affect early adhesion to abiotic surfaces. However, biofilm development is a multistep process that involves not only adhesion but also cell-cell interactions, matrix production, and morphological transitions. Altered glycosylation might trigger a compensatory response that upregulates genes involved in biofilm maturation, such as those related to ECP synthesis, or stress adaptation pathways (e.g., MAPK or HOG pathways), promoting a more robust biofilm structure despite reduced initial adherence.

Furthermore, remodeling of the cell wall can influence surface hydrophobicity and signaling pathways that enhance biofilm formation, as reported in other pathogenic fungi (https://www.mdpi.com/2309-608X/9/10/955#:~:text=Candida%20yeast%20can%20form%20a,stage%20of%20adhered%20cell%20filamentation.

https://www.frontiersin.org/journals/microbiology/articles/10.3389/fmicb.2021.638609/full

https://pmc.ncbi.nlm.nih.gov/articles/PMC95423/ ).

  1. Is the survival of MNT1/PMT2 silencer in the host but reduced pathogenicity due to increased sensitivity to oxidative killing (e.g., ROS) in the host? The survival rate of the silenced strains under the conditions of active oxygen production by H₂O₂ or macrophages should be measured, and the expression of antioxidant genes (such as SOD and CAT) should be analyzed to determine whether the expression is regulated by glycosylation.

Reply: Although we did not directly assess the susceptibility of the silenced strains to oxidative stress (e.g., HO exposure or ROS-specific assays), we did evaluate several other parameters that are highly informative in the context of the host-pathogen interaction. Specifically, we performed phagocytosis assays using macrophages and assessed additional indicators of immune recognition and fungal behavior during interaction with host cells (see Figure 8,10, and Table 4). These experiments provided valuable insights into the impact of MNT1 and PMT2 silencing on fungal virulence and immune evasion. The reduced virulence observed in the Galleria mellonella model (Figure 10) is consistent with these findings. While the contribution of oxidative stress sensitivity cannot be ruled out, our current data already supports a significant role of these genes in modulating fungal virulence. We agree that exploring the regulation of antioxidant defenses and ROS susceptibility would be an interesting avenue for future research. The host-pathogen interaction is such a complex relation that needs a thorough analysis, not only of the aspects already studied in our manuscript, and those suggested by the Reviewers. Phenomena such as immune sensing evaluation, nutritional stress, adhesion vs invasion vs cell damage, dynamics of fungal dimorphism in vivo, among others, are required to have a profound view of the interaction of this fungal species with the host. Having said that, we truly think that our dataset is robust enough to support our conclusions.

Round 2

Reviewer 1 Report

The criticisms were sufficiently addressed.

The criticisms were sufficiently addressed.

Author Response

No comment to address was found. We thank the Reviewer for his/her positive opinion.

Reviewer 3 Report

Manuela Gómez-Gaviria et al report the silencing of two genes involved in the S. schenckii O-linked glycosylation pathway and evaluate its contribution to the biology of this organism and its role in pathogen-host interaction. The silencing of MNT1 and PMT2 affected the interaction of S. schenckii with human PBMCs, macrophages, and neutrophils, along with its virulence. 

Although the technical methods used in this article have limitations, such as the difficulty of gene knockout and the lack of omics data exploration, gene silencing can provide evidence support to a certain extent. The second comments in last report was "Is the growth ability of silenced mutants in Figure 10 affected? Does this affect their virulence against galleria mellonella?". Sorry for the mistake that make you confused.

Author Response

We thank the Reviewer for commenting and clarifying his/her previous report. The growth rate of silenced strains was negatively affected in the MNT1 and PMT2-silenced strains, as reported in lines 446-449 of our manuscript. However, a similar fungal burden was observed in WT, control, and silenced strains, suggesting no defect in growth rates in the in vivo setting (see Table 4). This indicates that virulence attenuation is not because of a defect in adaptation to the host environment. Instead, it links proper protein glycosylation with virulence factors. This, along with explanations for the contrast in our in vitro and in vivo data, is part of the discussion section (see lines 880-897). So, no action was taken in the manuscript. We hope this clarifies your doubt.